# Hereditary Breast Cancer in Romania—Molecular Particularities and Genetic Counseling Challenges in an Eastern European Country

**DOI:** 10.3390/biomedicines11051386

**Published:** 2023-05-08

**Authors:** Andreea Cătană, Adrian P. Trifa, Patriciu A. Achimas-Cadariu, Gabriela Bolba-Morar, Carmen Lisencu, Eniko Kutasi, Vlad F. Chelaru, Maximilian Muntean, Daniela L. Martin, Nicoleta Z. Antone, Bogdan Fetica, Florina Pop, Mariela S. Militaru

**Affiliations:** 1Department of Molecular Sciences, Discipline of Medical Genetics, University of Medicine and Pharmacy Iuliu Hațieganu, Victor Babeș 8, 400347 Cluj-Napoca, Romania; 2Breast Cancer Tumour Center, Institute of Oncology I. Chiricuță, Republicii Nr. 34-36, 400015 Cluj-Napoca, Romania; 3Discipline of Medical Genetics, University of Medicine and Pharmacy Victor Babeș, Eftimie Murgu 2, 300041 Timișoara, Romania; 4Department of Oncology, Discipline of Surgery and Gynecological Oncology, University of Medicine and Pharmacy Iuliu Hațieganu, Republicii Nr. 34-36, 400015 Cluj-Napoca, Romania

**Keywords:** hereditary breast cancer, Romania, *BRCA*, non-*BRCA*

## Abstract

In Romania, breast cancer (BC) is the most common malignancy in women. However, there is limited data on the prevalence of predisposing germline mutations in the population in the era of precision medicine, where molecular testing has become an indispensable tool in cancer diagnosis, prognosis, and therapeutics. Therefore, we conducted a retrospective study to determine the prevalence, mutational spectrum, and histopathological prediction factors for hereditary breast cancer (HBC) in Romania. A cohort of 411 women diagnosed with BC selected upon NCCN v.1.2020 guidelines underwent an 84-gene NGS-based panel testing for breast cancer risk assessment during 2018–2022 in the Department of Oncogenetics of the Oncological Institute of Cluj-Napoca, Romania. A total of 135 (33%) patients presented pathogenic mutations in 19 genes. The prevalence of genetic variants was determined, and demographic and clinicopathological characteristics were analyzed. We observed differences among *BRCA* and non-*BRCA* carriers regarding family history of cancer, age of onset, and histopathological subtypes. Triple-negative (TN) tumors were more often *BRCA1* positive, unlike *BRCA2* positive tumors, which were more often the Luminal B subtype. The most frequent non-*BRCA* mutations were found in *CHEK2*, *ATM*, and *PALB2*, and several recurrent variants were identified for each gene. Unlike other European countries, germline testing for HBC is still limited due to the high costs and is not covered by the National Health System (NSH), thus leading to significant discrepancies related to the screening and prophylaxis of cancer.

## 1. Introduction

In Romania, 12,000 new BC cases are diagnosed annually, accounting for the second cause of cancer-related deaths after lung cancer [1]. Out of all diagnosed cases, 10% of BC are hereditary (HBC), the consequence of inherited or de novo predisposing mutations that define a group with increased malignancy risk compared to the general population. For the 11 million women in Romania, we do not have epidemiological data on the frequency of predisposing mutations in the general population. In addition, there are no national screening programs or reimbursed genetic testing. Nevertheless, more than 55% of cases are diagnosed in advanced clinical stages, with an overall survival rate below other European countries, defining a significant health concern [2]. *BRCA1* and *BRCA2* germline mutations are responsible for about 60% of HBCs with an overall 60–80% lifetime risk; the other 40% are associated with other predisposing variants in moderate-to-high penetrance genes such as *PALB2*, *PTEN*, *TP53*, *CDH1*, *CHEK2*, *ATM*, and the MMR group [3,4]. Although *BRCA*-related HBCs are prevalent, current testing guidelines recommend panel testing to ensure extensive mutation spectrum coverage [5]. Finding the predisposing pathogenic variants is essential for identifying high-risk women in order for them to be further followed in intensive screening programs that allow an early diagnosis or even avoid the onset of the malignancy and provide proper genetic counseling for family members [6,7]. In addition, the current medical practice supports targeted molecular therapy with PARP inhibitors among women with advanced breast neoplasia and triple-negative histology who have *BRCA* germline mutations and precision treatment [8]. We assume that the incidence of HBC in Romania is the same as that reported in the Caucasian population; however, it is already well-known that there are differences regarding the distribution of pathogenic variants and associations with the malignant phenotype [9,10]. There is very little data related to the distribution and peculiarities of germline variants in women from Romania, as published data only refer to 500 patients [11,12,13].

## 2. Materials and Methods

The study was conducted in the Department of Oncogenetics of the Oncology Institute of Cluj-Napoca, Romania, between 2018 and 2022. It included patients with at least one NCCN v.2020 molecular testing criteria for HBC susceptibility genes (BC diagnosed before age 50, bilateral BC metachronous or synchronous BC, TNBC before age 60). Based on the initial genetic consult, signing the consent form, and individual financial possibilities, eligible patients were tested using extensive molecular panels in two certified NGS/MLPA laboratories (Invitae and Blueprint) on a panel of 84 high-, moderate-, and low-penetrance genes (*AIP*, *ALK*, *APC*, *ATM*, *AXIN2*, *BAP1*, *BARD1*, *BLM*, *BMPR1A*, *BRCA1*, *BRCA2*, *BRIP1*, *CASR*, *CDC73*, *CDH1*, *CDK4*, *CDKN1B*, *CDKN1C*, *CDKN2A*, *CEBPA*, *CHEK2*, *CTNNA1*, *DICER1*, *DIS3L2*, *EGFR*, *EPCAM*, *FH*, *FLCN*, *GATA2*, *GPC3*, *GREM1*, *HOXB13*, *HRAS*, *KIT*, *MAX*, *MEN1*, *MET*, *MITF*, *MLH1*, *MSH2*, *MSH3* MSH6*, *MUTYH*, *NBN*, *NF1*, *NF2*, *NTHL1*, *PALB2*, *PDGFRA*, *PHOX2B*, *PMS2*, *POLD1*, *POLE*, *POT1*, *PRKAR1A PTCH1*, *PTEN*, *RAD50*, *RAD51C*, *RAD51D*, *RB1*, *RECQL4*, *RET*, *RUNX1*, *SDHA*, *SDHAF2*, *SDHB*, *SDHC*, *SDHD*, *SMAD4*, *SMARCA4*, *SMARCB1*, *SMARCE1*, *STK11*, *SUFU*, *TERC*, *TERT*, *TMEM127*, *TP53*, *TSC1*, *TSC2*, *VHL*, *WRN*, *WT1*).

Genomic DNA obtained from peripheral blood samples was enriched for targeted regions using a hybridization-based protocol and sequenced using Illumina technology. All targeted regions were sequenced with ≥50× depth and 20 bp of flanking intronic sequence, Reads were aligned to a reference sequence (GRCh37), and sequence changes were identified and interpreted in the context of a single clinically relevant transcript. Promoters, untranslated regions, and other non-coding regions were not interrogated.

Only patients with pathogenic and potentially pathogenic mutations were included in the study, although 76 patients presented variants of uncertain significance (VUS) but were not included in the positive group statistical analysis. Data were organized using Microsoft Excel, part of the Microsoft Office 2019 suite (Microsoft Corp., Redmond, WA, USA). Data were then analyzed using R 4.2.2 (R Foundation) [14], RStudio (Posit Software, PBC, Boston, MA, USA) [15]. In addition, the following libraries were loaded in the workspace: stringr [16], readxl [17], and GenVisR [18]. To identify statistically significant differences in quantitative variables between groups, we used the Wilcoxon rank-sum (Mann–Whitney U) test. To identify statistically significant differences in frequencies of qualitative variables between groups, we used either the Chi^2^ test, or where its assumptions were violated, the Fisher test. Quantitative variables were expressed as mean (standard deviation), and qualitative variables were expressed as percentages.

## 3. Results

Of 624 women diagnosed with BC, 560 met at least one NCCN v.2020 molecular testing criteria, and 411 women underwent testing.

Among all 411 patients, 135 (32.8%) carried a pathogenic or likely pathogenic heterozygous germline mutation in 19 genes, including high-penetrant breast cancer genes like *BRCA1* (43–31.9%), *BRCA2* (19–14.1%), *PALB2* (12–8.9%), *TP53* (3–2.2%), and CDH1 (2–1.5%), and moderate to low-penetrant genes *CHEK2* (25–18.5%), *ATM* (12–8.9%), MMR group *PMS2* (2–1.5%), *MSH3* (1–0.7%), *MSH6* (1–0.7%), *MLH1* (1–0.7%), *BARD1* (3–2.2%), *NF1* (3–2.2%), *MUTYH* (5–3.7%), *EGFR* (1–0.7%), *SDHB* (1–0.7%), *RAD50* (3–2.2%), *NBN* (2–1.5%), and *XRCC2* (2–1.5%). We identified 142 defects in all. Of these, 77 were different pathogenic variants, of which the most common defects were frameshift (46–32.4% of all detected defects) and missense (40–28.2%) variants, followed by nonsense (27–19%), deletion/insertion (21–14.8%), and intronic variants (8–5.6%). Seven patients in the group had two pathogenic mutations (Figure 1 and Figure 2). 

Three recurrent variants (reported in more than three unrelated patients) c.3607C>T (p.Arg1203Ter), c.181T>G (p.Cys61Gly), and c.5266dupC (p.Gln1756Profs) accounted for (26) 60% of all reported variants were identified in the *BRCA1* gene. The c.172_175delTTGT (p.Gln60Argfs) frameshift variant was also recurrent in the *PALB2* gene, reported in (4) 33% of carriers, and c.1564_1565delGA(p.Glu522Ilefs) frameshift variant was found in (5) 41% of *ATM* carriers. Another recurring variant in the CHEK2 gene, c.470T>C (p.Ile157Thr), was reported, accounting for a majority of (18) 72% of the pathogenic variants for this gene and 13% of all non-*BRCA* pathogenic variants in this cohort (Table 1).

The mean age at the primary cancer diagnosis was 41.387 ± 8.084 for *BRCA* carriers; the mean age at primary cancer diagnosis was significantly higher both for non-*BRCA* mutation carriers (at 44.466 ± 7.02, Wilcoxon rank sum *p* = 0.007) and for patients with no mutation (at 45.62 ± 7.367, Wilcoxon rank sum *p* < 0.001).

The mean age at diagnosis for patients with a positive family history of cancer was 42.798 ± 7.054 versus 45.674 ± 7.607 for patients with no family history of cancer (Wilcoxon rank sum *p* = 0.002). Among patients with no mutation, 24.5% had positive familial history. In contrast, among patients with any mutation, 49.2% had a positive familial history (54.2% among those with BRCA mutations and 45.2% among those with other types of mutation). There were statistically significant differences between patients with no mutations and both those with BRCA mutations (Chi^2^
*p* < 0.001) and those with non-BRCA mutations (Chi^2^
*p* < 0.001). At the same time, there were no differences among *BRCA* and non-*BRCA* carriers (Chi^2^
*p* = 0.302). We did not have information about the family history of 3 patients with BRCA mutations.

The most common tumor histology in the cohort was invasive ductal carcinoma, found in 312 of all tested patients (75.9%), followed by invasive lobular carcinoma, 53 (12.9%), in-situ carcinoma, 30 (7.3%), and other rare histologies, 16 (3.9%). As for the germline-positive patients, the most common tumor histology was invasive ductal carci-noma 102 (75.5%), followed by invasive lobular carcinoma 13 (9.6%) and other rare his-tologies 11 (8.1%). In situ, non-invasive histologies were identified in 9 (6.7%) of diagnosed cases. There were statistically significant differences in the prevalence of tumoral histology types across the mutation types (Fisher exact test *p* = 0.018). Rare histologies had a lower prevalence in patients with no mutation (1%) than in patients with mutations, while lobular histology was much rarer among patients with BRCA mutations than the others. An interesting finding was that the indication for genetic testing for 42% of patients without familial history of cancer was the TN histology revealing a *BRCA1* mutation. A total of 38% of Luminal A cancer patients with an onset <45 years of age and no familial history of cancer revealed *BRCA2* or non-*BRCA* germline variants (Table 2).

The most common molecular subtype for germline-positive patients was Luminal B (77–57%), followed by triple negative (33–24.4%) and Luminal A (25–18%) subtypes. Luminal B histology was predominant in patients with no mutation compared to mutation carriers (Chi^2^
*p* < 0.001, OR = 3.68 CI 95% 2.23–6.08). TN histology was predominant among BRCA mutations, compared to the other groups (Chi^2^
*p* < 0.001, OR = 6.71 CI 95% 3.57–12.65). The distributions of the molecular types in breast cancer patients with the BRCA and non-BRCA mutations differed significantly depending on the gene involved. The Luminal A subtype was prevalent in tumors positive for moderate-to-low penetrance mutations. The *BRCA1*-associated cancers were significantly more often TN than tumors harboring other mutations (Chi^2^
*p* < 0.001, OR = 9.69 CI 95% 4.83–19.45). Luminal B subtypes, particularly Luminal B HER2-positive subtypes, were reported more frequently in *BRCA2* tumors but with no statistical significance compared to the other *BRCA1* and non-*BRCA* tumors (*p* = 0.158). The Luminal A subtype was more frequently associated with CHEK2-positive tumors than other mutations (*p* = 0.013).

The mean Ki67 index value was 45.194 ± 23.717 for *BRCA* carriers, significantly higher than for non-*BRCA* mutation carriers (35.808 ± 20.427, Wilcoxon rank sum *p* = 0.016) but not compared to the mean value of Ki67 index in patients with no mutations (42.175 ± 19.681, Wilcoxon rank sum *p* = 0.366) (Figure 3).

## 4. Discussion

### 4.1. Mutation Prevalence

In countries such as Romania, Bulgaria, Ukraine, Malta, Albania, Serbia, Bosnia, and Herzegovina, Macedonia, and Montenegro, surveys on *BRCA1* and *BRCA2* mutations have yet to be conducted, and no data are available. From Croatia, no conclusive data about the founder *BRCA1/2* mutation pattern is available, since only some individual mutations and benign variants were reported in one study [19].

It is complicated to estimate the prevalence of pathogenic mutations in the general population, considering that until now, there have only been two studies published about the Romanian population. One, published in 2022, studying 250 women with breast cancer and 240 with ovarian cancer who underwent germline molecular testing for the detection of pathogenic *BRCA1* and *BRCA2* mutations, revealed that the most common variants identified were 5266delC, followed by 4218delG and c.68_69delAG for *BRCA1*, and c.9371A>T and c.1528G>T for *BRCA2* [11]. The other study conducted on 130 breast cancer patients tested by multigene panel analysis (*BRCA1*, *BRCA2*, *TP53*, *STK11*, *CDH1*, *PTEN*, *PALB2*, *CHEK2*, *ATM*), highlighted *BRCA1* c.3607C>T as the most common variant in the group, prevalent in triple-negative invasive carcinomas [12].

As expected, the most common mutations in our group were found in *BRCA1* and *BRCA2* genes, exceeding half of the reported mutations in this study. Frameshift and missense mutations leading to a complete or partial loss of tumor suppressor effect in *BRCA* genes were the most common defects. The c.3607C>T, c.181T>G missense variants, and c.5266dupC and c.68_69delAG(p.GluValfs) frameshift variant were recurrent in *BRCA1* carriers. The c.181T>G is one of the most common causes of breast and ovarian cancer in patients with Eastern European and Polish heritage and in Sicily [20,21,22], and therefore expected to be a common variant in our study. An older study in a Nord-Eastern region of Romania revealed that the BRCA1 5382insC mutation was not observed in any of the 120 breast and 50 ovarian cancer patients, contradictory when compared to reported data for the Romanian and Eastern European populations [23].

Interestingly, only one *BRCA2* mutation was found in 3 patients, the c.9371A>T (p.Asn3124Ile) missense variant, a founder defect in the eastern European population.

Only three other studies in the same demographics account for 320 patients with pathogenic mutations related to breast cancer. One study including 107 patients diagnosed with breast or ovarian cancer revealed that the c.5266dupC Ashkenazi founder mutation was the most common *BRCA1* pathogenic variant reported in 36.67% of cases, closely followed by c.3607C>T in 30% of cases. Only one *BRCA2* variant, c.9371A>T, was recurrent in this study [11]. Another smaller study, including 44 breast cancer patients revealed the same pattern in the *BRCA1* mutational prevalence; *BRCA1* c.5266dupC and c.3607C>T, *BRCA2* c.9371A>T were also prevalent in this group [13]. The previous study, including 56 patients, revealed a higher prevalence for *BRCA1* c.3607C>T variant than c.5266dupC and a recurrent *BRCA2* mutation, c.8755-1G>A [12].

Mutations in the *PALB2* gene, a high-penetrant gene associated with a 40–60% lifetime risk of BC, were reported as the third most common high-risk gene in this study. One particular variant, c.172_175delTTGT (p.Gln60Argfs), was recurrent in *PALB2* carriers and was reported in other studies to be associated with hereditary breast and pancreatic cancer [24,25]. There are no available data on *PALB2* mutation prevalence in the Romanian population. Only one study included East European and, thus, Romanian cancer-diagnosed patients, but with no detailed description of *PALB2* mutation [26]. With a frequency ranging from 0.5 to 1.0%, the truncating *PALB2* variant c.172_175delTTGT has recently been discussed in Central and Eastern Europe [27], Poland, Belarus, Germany, and Russia [28,29,30]. Considering the aforementioned, c.172_175delTTGT could be considered a PALB2 founder mutation for the Romanian population. Another mutation, the c.2257C>T (p.Arg753Ter), also seems to be recurrent for the *PALB2* gene and has previously been reported in Poland and the Eastern-European population [31].

Germline testing for breast cancer should also include the PALB2 gene, considering the frequency of pathogenic defects in the general population and because *PALB2* heterozygotes should be considered for the same therapeutic regimens and clinical trials as those for BRCA1 and BRCA2 carriers.

*ATM* is a moderately penetrant gene associated with a 20–40% risk for breast cancer. The recurrent Slavic founder mutation c.1564_1565delGA (p.Glu522Ilefs) frameshift variant was found in 41% of *ATM* carriers and previously reported in the Romanian population [14]. 

Consistent with currently available data, pathogenic variants in *CHEK2* were the most frequently identified after pathogenic variants in *BRCA1* or *BRCA2* genes. Debates on the impact of *CHEK2* pathogenic mutations on breast cancer risk are ongoing, with emerging data to classify pathogenic mutations of *CHEK2* in moderate to low-penetrant variants [32]. The c.470T>C (p.Ile157Thr) variant was recurrent in our cohort, with more than half of *CHEK2* mutated patients carrying this mutation. Compared to other *CHEK2* pathogenic variants, c.470T>C has an attenuated association with BC, was not associated with non-breast cancers [33,34], and is probably modulated by other genetic factors or non-genetic risk factors to increase BC risk. Along with hormone-related risk factors [35], it has been shown that a family history of BC correlates with higher risks for women with *CHEK2* pathogenic variants [36,37,38]. Surprisingly, the common c.1100del *CHEK2* variant was reported only in one patient. On the other hand, two out of three patients carrying the c.902delT *CHEK2* moderate-risk variant had other germline pathogenic mutations in the *BRCA1* gene.

One unexpected finding in our study is related to the *MUTYH* gene. The association between *MUTYH* mutations and HBC risk is controversial, as there a higher level of evidence that carriers homozygous for MUTYH pathogenic variants have an increased risk of BC [39]. However, a higher frequency of heterozygous *MUTYH* mutations in families with breast and colorectal cancer has also been reported compared to the general population [40,41]. We identified the c.650G>A recurrent heterozygous mutation in three patients in our cohort. Two patients had invasive lobular carcinoma and intestinal polyps, and the third had a parent diagnosed with colon cancer. Among other variants, the c.650G>A mutation is reported in ClinVar to be associated with invasive Luminal B breast carcinoma [42].

### 4.2. Molecular Subtypes Associations

Invasive carcinoma of the breast is considered a heterogeneous group of malignant epithelial tumors. Current data describes a wide range of morphological phenotypes and specific histopathological and molecular subtypes among sporadic and hereditary types but with significant differences between genotypes of germline mutations-associated tumors [43,44]. Most *BRCA1*-associated breast cancers are invasive ductal carcinomas of non-special type and fall into the “basal-like” intrinsic molecular subtype [45]. These triple-negative (TN) tumors lack estrogen, progesterone receptors, and human epidermal growth factor receptor 2 expressions. *BRCA2*-associated breast cancers are more likely to be found in the “luminal type” and share common characteristics with sporadic tumors [46]. Luminal A tumors have Ki67 ≤ 14% and lack HER2 protein overexpression. Luminal B tumors have higher Ki67 values and are HER2-positive [47]. In our study, basal histology was a common finding in BRCA1 patients, and luminal types were more commonly identified in BRCA2 and non-BRCA carriers. Ductal in situ carcinoma (DISC) was a rare histological finding in our cohort as most diagnosed cases were symptomatic and therefore T2/T3, N/M > 1 at diagnosis. Less than 5% of cases were diagnosed as DISC, all part of annual screening due to a positive family history of breast cancer.

Ki-67 proliferative marker is considered an essential prognostic in breast cancer. It has a significantly higher expression in *BRCA*-positive breast cancer [48,49]. It implies its potential as an efficient prognostic factor in *BRCA*-positive breast cancer and future therapeutic implications in the context of emerging data suggest it might be advantageous to promote, rather than hinder, cell proliferation for immunotherapy to be optimally effective [50,51].

### 4.3. Cohort Particularities

We consider that our study group is representative of Romania since the women enrolled came from different geographic areas and ethnicities (multidimensional scaling analysis supported the genetic similarity of the Wallachia, Moldavia, and Dobrudja groups with the Balkans, while the Transylvanian population was closely related to Central European groups) [52].

An expected finding was that we had a higher mutation prevalence than other studies. Most patients addressing genetic counseling services had at least two or three NCCN-based testing criteria. We noticed that most patients had at least two or more NCCN-based eligibility criteria for genetic testing. This particularity implicitly associates a higher positivity rate than other groups where patients were tested with only one criterion. The over-selection is mainly due to inappropriate screening and testing guidelines in hereditary cancers. Among the factors contributing to the underutilization of genetic testing services in Romanians, we mention lower awareness of testing among patients and medical staff and support for obtaining genetic counseling and testing, particularly in resource-limited settings representing 60% of the general population. Genetic counseling, testing, and after-testing discussions are often described as complicated and inaccessible for many women in Romania. Paradoxically, approximately 40% of patients with eligibility testing criteria did not have the genetic test because of objective financial impediments. Genetic testing in Romania must be supported by the government or reimbursed by health insurance to avoid significant discrepancies between socio-economic groups.

Recurrent mutations in our study were c.3607C>T, c.181T>G, c.5266dupC in the *BRCA1* gene, c.9371A>T in the *BRCA2* gene, c.172_175delTTGT in the *PALB2* gene, c.1564_1565delGA in the *ATM* gene, c.470T>C in *CHEK2* gene, and c.650G>A in the *MUTYH* gene. For high-penetrant genes such as *BRCA1*, *BRCA2*, *PALB2*, and moderate-high penetrant genes such as *ATM*, recurrent mutation frequency converges to already published data for Eastern European populations. We observed a high frequency for *CHEK2* variants, the most frequent moderate-risk breast cancer predisposition gene. Despite contradictory data on the c.407T>C pathogenicity, this variant may have more than a polygenic role model in breast carcinogenesis and deserves further large-data analysis. An unexpected finding in our cohort was the presence of heterozygous *MUTYH* pathogenic variants, among which c.650G>A was recurrent and already in the attention of various investigators for the association with invasive breast carcinomas [41].

We also evaluated the attitude regarding surgical prophylaxis among women with a high risk of bilateralization. If most patients with pathogenic mutations in genes with increased penetrance opted for prophylactic mastectomy in favor of conservative imaging methods, paradoxically, more patients with pathogenic mutations in genes with low or moderate penetrance requested genetic or surgical consult to perform radical surgical prophylaxis. In this regard, prophylactic mastectomy recommendations were always preceded by the Tumour Board assessment and psychological counseling to avoid unnecessary interventions.

Genetic counseling has been challenging due to insufficient genetic screening programs and medical education. For example, an extensive pre-pandemic report revealed that even in developed countries, 50% of women diagnosed with breast cancer do not receive genetic counseling [53]. If genetic counseling was available without significant impediments for educated patients younger than 40 years of age, in older women or women with low education, we confronted issues related to understanding the information, advantages, and the medical use of genetic testing. In this case, more than one genetic counseling session or integration of another family member was necessary. Since most of our subjects were submitted to genetic testing at diagnosis, a critical timing in patients’ medical management also focused on identifying subjects at higher risk of psychological distress to address them for psychological support and give them the appropriate coping strategy. We observed that a positive genetic test still creates a significant emotional stigma for the patient and other family members and therefore requires professional assistance to avoid further emotional distress.

## 5. Conclusions

The genetic characteristics of HBC in Romania are similar to those reported for the East Caucasian and Slavic populations. *BRCA* carriers and patients with a family history of cancer are diagnosed with breast cancer earlier than carriers of other mutations. TN tumors are associated with *BRCA1* mutations, while Luminal B subtype tumors are associated with *BRCA2* mutations. Further attention is recommended for moderate-penetrant genes like *CHEK2* to determine the role of pathogenic variants in assessing BC predisposition. Romania, part of the EU, requires the implementation of genetic screening programs and proper genetic counseling services to reduce the number of women with hereditary genetic components and late diagnosis.

## Figures and Tables

**Figure 1 biomedicines-11-01386-f001:**
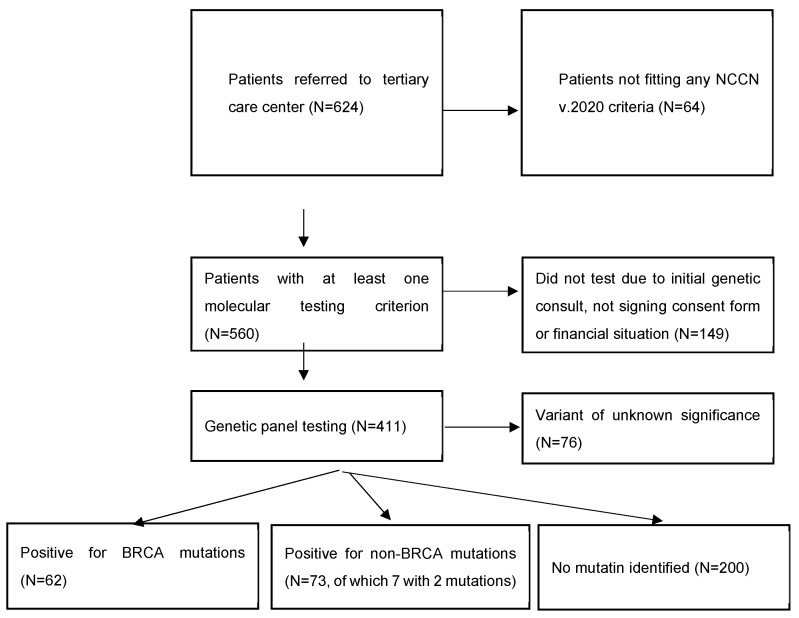
General characteristics of the cohort.

**Figure 2 biomedicines-11-01386-f002:**
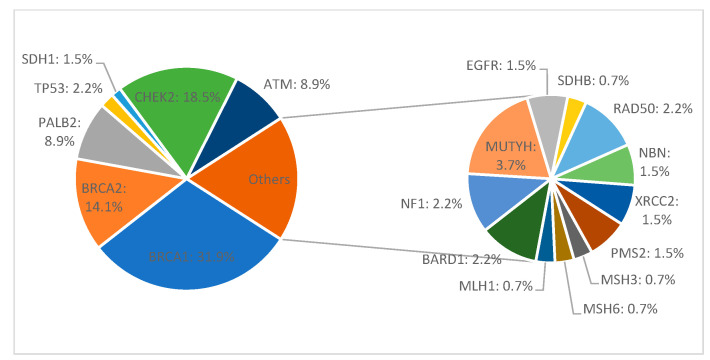
Mutation frequency in germline-positive patients with hereditary breast cancer.

**Figure 3 biomedicines-11-01386-f003:**
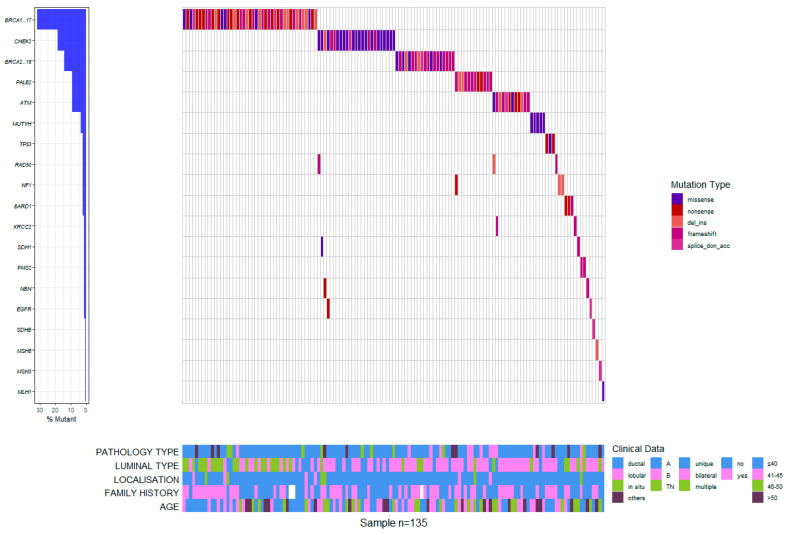
Waterfall plot of mutation profiles and clinical data for the Romanian cohort.

**Table 1 biomedicines-11-01386-t001:** Pathogenic variants reported in breast cancer patients.

Gene	Mutation	Number of Patients
** *BRCA1* **	c.3607 C>T (p.Arg1203Ter)	12
	c.181T>G (p.Cys61Gly)	8
	c.5266dupC (p.Gln1756Profs)	6
	c.68_69delAG (p.GluValfs)	3
	c.1687C˃T (p.Glu563Ter)	2
	c.843_846del (p.Ser282fs)	1
	c.212+1G>T	1
	c.4327C>T (p.Arg1443Ter)	1
	c.5030_5033delCTAA (p.Thr1677Ilefs)	1
	c.1018C>T (p.Arg340Ter)	1
	c.4065_4068del (p.Asn1355fs)	1
	c.3700_3704del (p.Val1234fs)	1
	c.737del (p.Asp245_Leu246isTer)	1
	c.5251C>T (p.Arg1751Ter)	1
	c.213-12A>G	1
	c.1636_1654del (p.Met546fs)	1
	c.211A>G (p.Arg71Gly)	1
** *BRCA2* **	c.9371A>T (p.Asn3124Ile)	3
	c.2808_2811del (p.Ala938Profs)	2
	c.5796_5797del (p.His1932fs)	2
	c.7878G>C (p.Trp2626Cys)	2
	c.2944A>C (p.Ile982Leu)	1
	c.7230delT (p.Phe2410Leufs)	1
	c.9253delA (p.Thr3085Glnfs)	1
	c.3545_3546delT (p.Phe1182Terfs)	1
	c.793+1G>A	1
	c.5576_5579del (p.Ile1859fs)	1
	c.3680_3681del (p.Leu1227fs)	1
	c.8680C>T (p.Gln2894Ter)	1
	c.729_732del (p.Asn243fs)	1
	c.5946del (p.Ser1982fs)	1
** *PALB2* **	c.172_175delTTGT (p.Gln60Agfs)	4
	c.2257C>T (p.Arg753Ter)	3
	c.93dupA (p.Leu32Thrfs)	1
	c.757_758del (p.Leu253fs)	1
	c.1037_1041del (p.Lys346fs)	1
	c.93dup (p.Leu32fs)	1
	c.1002C>A (p.Tyr334Ter)	1
** *TP53* **	C.586C>T (P.Arg196ter)	1
	c.1025G>C (p.Arg342Pro)	1
	c.916C>T (p.Arg306Ter)	1
** *SDH1* **	c.1531C>T (p.Gln511Ter)	2
** *ATM* **	c.1564_1565delGA (p.Glu522Ilefs)	5
	c.935dup (p.Leu312Phef*s6)	1
	c.6095G>A (p.Arg2032Lys)	1
	c.8585-2A>C	1
	c.5980A>T (p.Lys1994Ter)	1
	c.4768C>T (p.Leu1590Phe)	1
	c.5644C>T (p.Arg1882Ter)	1
	c.5932G>T (p.Glu1978Ter)	1
** *CHEK2* **	c.470T>C (p.Ile157Thr)	18
	c.902delT, p.(Leu301Trpfs*3)	3
	c.917G>C (p.Gly306Ala)	1
	c.1312G>T (p.Asp438Tyr)	1
	c.444+1G>A	1
	c.1100del (p.Thr367fs)	1
** *BARD1* **	c.1690C>T (p.Gln564Ter)	1
	c.632T>A (p.Leu211Ter)	1
	c.176_177del (p.Glu59fs)	1
** *RAD50* **	c.326_329del (p.Thr109fs)	3
** *PMS2* **	c.1239dup (p.Asp414fs)	1
	c.1076dupT(p.Leu359Phefs)	1
** *MSH3* **	c.2436-1G>A	1
** *MSH6* **	ex.1-6del	1
** *MLH1* **	c.544A>G (p.Arg182Gly)	1
** *MUTYH* **	c.650G>A (p.Arg217His)	3
	c.1187G>A (p.Gly396Asp)	1
	c.536A>G (p.Tyr179Cys)	1
** *NF1* **	ex.5 CNV	1
	ex.7del	1
	c.2410-1G>A	1
** *NBN* **	c.657_661del (p.Lys219fs)	2
** *SDHB* **	c.423+1G>A	1
** *XRCC2* **	c.190C>T (p.Arg64Ter)	2
** *EGFR* **	c.2061+2T>C	2

**Table 2 biomedicines-11-01386-t002:** Histopathological and molecular characteristics for the cohort of 411 patients. ^†^ = 3 patients with BRCA mutations lacked information regarding family history of breast cancer and were excluded from analysis.

Variable	BRCA Mutation (N = 62)	Non-BRCA Mutation (N = 73)	VUS (N = 76)	No Mutation (N = 200)
Diagnosis age	41.387 ± 8.084	44.466 ± 7.02	44.7	45.62 ± 7.367
Positive family history of cancer	32 (54.2%) ^†^	33 (45.2%)	32 (42.1%)	49 (24.5%)
Histology				
Ductal (312–75.9%)	50 (80.6%)	52 (71.2%)	48 (63.1%)	162 (81%)
In situ 30 (30–7.3%)	4 (6.5%)	5 (6.8%)	11 (14.4%)	10 (5%)
Lobular (53–12.9%)	4 (6.5%)	9 (12.3%)	14 (18.4%)	26 (13%)
Others (16–3.9%)	4 (6.5%)	7 (9.6%)	3 (4.1%)	2 (1%)
Molecular subtype				
Luminal A (72–17.5%)	10 (16.1%)	15 (20.5%)	28 (36.8%)	19 (9.5%)
Luminal B (285–69.3%)	28 (45.2%)	49 (67.1%)	42 (55.2%)	166 (83%)
TN (54–13.1%)	24 (38.7%)	9 (12.3%)	6 (7.9%)	15 (7.5%)

## Data Availability

Data is available on request to the corresponding author.

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
