# Peer review of "Hereditary Breast Cancer in Romania—Molecular Particularities and Genetic Counseling Challenges in an Eastern European Country"

_biomedicines, 2023, doi:10.3390/biomedicines11051386_

Round 1

Reviewer 1 Report

The rationale for this project is based on paucity of information concerning the prevalence of inherited pathogenic variants in breast cancer predisposition genes, such as BRCA1 and BRCA2 in breast cancer cases from  Romania. It is well known that carrier frequency may vary according to the population studied and this information could inform genetic testing and counselling practices.

This retrospective study investigated 411 women diagnosed with breast cancer from 20181 to 2022 in one Romanian clinic. Selection of cases  for genetic testing was based on well recognized NCCN (American)guidelines . Although interest was aimed at determining the contribution of known breast cancer predisposing genes, an 84-gene panel was investigated using NGS and MLPA methods where appropriate. The findings are interesting as pathogenic variants were identified in BRCA1, BRCA2, PALB2, TP53, ATM and CHEK2 (high to low/moderate risk genes), some of which were found to recur in this population. The results are presented and discussed in light of clinical metrics. The findings provide further insight in the prevalence of pathogenic variants in breast cancer predisposing genes in Romanians with breast cancer and add to the growing knowledge of the relevance of such genes in breast cancer regardless of country of origin of cancer cases.

However, to properly evaluate the relevance of findings and relate them to similar published data from other populations the data could be presented in a systematic way as follows:

1) present typical clinical metrics data (histological type of breast cancer, age diagnosis, family history, etc) of 411 cases in table format which also includes data concerning carrier frequencies for major genes (BRCA1, BRCA2, PALB2, TP53, etc) and other genes (RAD50, etc) so that statistical analyses can be verified. The data used to calculate percentages and significance of findings  is missing from the manuscript as only percentages are provided.

Figure 2 which contains some of this data is unreadable and data within cannot be interpreted without data from mutation-negative  carriers.

2) provide clinical metrics (histology, age diagnosis, etc) for each pathogenic variant carrier shown in Table 1 in supplementary file. This could help researchers and clinicians identify associations. 

3) Some of the molecular subtype association data presented in Discussion section 4.2 should appear in the Results section along with actual data not only percentages. 

4) The data in the cohort particularities section of the Discussion should also appear in the Results section and include findings (ie data not only percentages).

5) Methods section is missing a description of the criteria used to describe the clinic metrics and family history for the cohort.

6) Methods section should provide NCCN criteria used to select cases rather only the version.

7) Methods section is missing description of criteria used to investigate "cohort particularities".

8) The full list of 84 genes tested should be provided as this information is useful to readers along with classification of penetrance for breast cancer as a small number of genes have been validated in increasing risk to breast cancer. 

9) In methods section provide criteria used to evaluate classification of variant as pathogenic from VUS or benign.  The assumption is that all variants identified in Table 1 were present in heterozygous state in carriers, including those in MUTYH? This fact should be clarified in study design.

10) Figure 2 is unreadable. 

Other comments:

1) include references to the two Romanian studies and information concerning them in the introduction section of the study to rationalize the present study (Page 2). 

2) Correct comment on Page 2 lines 46-47. While 60% of germline BRCA1 or BRCA2 pathogenic variants may be identified HBCs depending on population studied, the remaining 40% are not accounted by the genes listed. Indeed,  there remains HBCs not accounted for known genes. Also, it is important to state that the proportions of carriers and involvement of known genes may differ depending on population studied and inclusion criteria of defining HBC/familial breast cancer. 

3) A number of general statements were made in the Introduction but not referenced (second half of the introductory paragraph) and this should be corrected.

4) References missing or misplaced for comments made concerning founder effect (Page 6, line 142); opening sentences on Page 6 line 143-144, 153-154, and 163-164 and Page 7 line 220-221.

5) typo to fix on Page 7 line 209 ("o" should be "to")

6) what "other studies" are being referred to (see Page 7 lines 236-237) as no references were provided for most of the comments made in the discussion section 4.3 regarding cohort particularities. 

Author Response

All changes were noted in the manuscript text by ‘Comments’

Reviewer 1

  • We introduced typical clinical metrics data (histological type of breast cancer, age diagnosis,
  • Family history, etc) of 411 cases in table format which also includes data concerning carrier frequencies for major genes (BRCA1, BRCA2, PALB2, TP53, etc) and other genes (RAD50, etc) so that statistical analyses can be verified.
  • Figures and tables were modified (we added data for controls=negative carriers, provided data on histology and molecular characteristics in a new table, added numbers to existing percentages.
  • Data presented in Discussion section 4.2 was moved to the Results section.
  • Methods section was completed with a description of the criteria used to describe the clinic metrics, family history for the cohort and NCCN criteria used to select cases.
  • We completed the data for "cohort particularities".
  • We added the full list of 84 genes tested and provided criteria used to evaluate classification of variant as pathogenic
  • MUTYH was clarified as heterozygous state for all patients.
  • We included references to the two Romanian studies and information concerning them in the introduction section of the study.
  • We added references for the general statements in the Introduction.

Reviewer 2 Report

The proposed article, entitled 'Hereditary Breast Cancer in Romania - Molecular Particularities and Genetic Counseling Challenges in an Eastern European Country", deals with a useful and original topic making the proposed article interesting and worthy of consideration.

The proposed article is easy to understand because it is well constructed, clear and well described with comprehensive figures appropriate to the subject matter.

In my opinion, this work is acceptable for publication after minor revisions that will help the authors to improve the quality of their manuscript.

- in the introduction please discuss the role of BRCA1/2 and the other high and moderate penetrance genes on breast cancer, the current recommendations for genetic testing, and the percentage risk of developing breast cancer for carriers of pathogenic variants in BRCA1/2;

- please use acronyms for words that repeat frequently throughout the text;

- in the results paragraph, please include a sub-section with the clinical characteristics of the patients. For the sake of clarity, it would also be better to add a table containing all the clinical-pathological data;

- please expand the paragraph on materials and methods (techniques, classification of variants, which 84 genes are analysed, etc.); 

- did you find any pathogenic variants particularly represented in any geographical area of Romania?;

- among the pathogenic variants found in your population, one in particular, c.181T>G, identified mainly in eastern European populations (Polish, Czech, Slovenian, Austrian, Hungarian, Belorussian and German), was also observed in a study carried out in sicily. Please cite and discuss the following work (doi: 10.1007/s13167-010-0037-y , 10.3390/cancers12051158); 

- please carefully revise the English language throughout the text and correct some grammatical errror and trivial imperfections.

Author Response

Reviewer 2

  • (PMID: 34371384) was added in the Introduction.
  • We used acronyms for words that repeat frequently throughout the text.
  • Figures and tables were modified (we added data for controls=negative carriers, provided data on histology and molecular characteristics in a new table, added numbers to existing percentages.
  • Methods section was completed with a description of the criteria used to describe the clinic metrics, family history for the cohort and NCCN criteria used to select cases
  • We completed the data for "cohort particularities".
  • We added the full list of 84 genes tested and provided criteria used to evaluate classification of variant as pathogenic
  • Data on c.181T>G was added.
  • We added (doi: 10.1007/s13167-010-0037-y) and Janavičius R. Founder BRCA1/2 mutations in the Europe: implications for hereditary breast-ovarian cancer prevention and control. EPMA J. 2010 Sep;1(3):397-412. doi: 10.1007/s13167-010-0037-y. Epub 2010 Jun 27. PMID: 23199084; PMCID: PMC3405339.

Round 2

Reviewer 1 Report

The revised manuscripts has been improved, especially with the introduction of the new Figure 1 and Table 2,  edits to the methodology (defining NCCN guidelines for genetic testing) and inclusion of genes. 

However, there remains significant issues with the statements made in the results section as do not always align with data presented in the tables 1 and 2.

Table 1 includes changes in the number of carriers of each variant identified but statements made concerning multiple carriers of the same variant not match the text as detailed below. 

Also, I could not follow a number of the statements made in the results section concerning data presented in Table 2. As only percentages are given in with some statements,  it is difficult to verify of they are correct. 

The overriding issues with the results section is that many of statements made in the results  section are not necessarily aligned with the data presented in Tables 1 and 2 and should be rechecked for accuracy. This could be rectified by providing the complete statements concerning the numbers used to calculate frequencies (ie X/Y, %) and statistical comparison of defined groups. Some of this information could also be added directly to the tables to reduce redundancy. 

Below I have provided some examples of the discordance of text with results in the Tables but other discrepancies exist and were too numerous to state in this review. 

Examples of numerical discrepancies: 

Line 138 and Figure 2 title: clarify that the prevalence of number of pathogenic or likely pathogenic germline mutations identified in 19 different genes is presented as a frequency of variants identified in each gene. For example, BRCA1 variants represent 31.9% or 43 of all 135 variants identified (rather than prevalence in 411 patients tested for BRCA1 variants). 

Lines 153- 154: recheck the data presented in Table 1 for BRCA1 variants as it does not match statements made in these lines. Indeed Table 1 has crossed out the number of c.3607C>T and c.68_69delAG carriers and now includes greater than 3 carriers of c.1887C>T. 

Lines: 153-161: recheck percentages given for totals of recurrent mutations for all of data presented in Table 1. For example the 18 CHEK2 c.470T>C  carriers account for 72% (or 18) of all 25 carriers of CHEK2 variants, not 60% as stated. 

Line 172-173: recheck percentages stated in this sentence as they do not match data in Table 2 and so it is difficult to determine if statistical analyses presented in lines 175-177 is correct.

Lines 179-220: I am unable to follow most of the statements made concerning frequency of histopathological types with respect to carrier status when cross referenced with data presented in Table 2.  For example there are 264 ductal carcinomas accounting for 78.8% of all 335 breast cancer cases categorized for histology – so where does “107” come from?  There are 39 lobular carcinomas accounting for 5.7% of all 335 histologically classified breast cancer – so where does n=16 and 11.6% come from? More examples of these inconsistencies in data occurs throughout this section. As mentioned above , this could be corrected by providing totals in the table and defining the numbers referred to in the comparative groups in the text (ie X/Y, %). In this way analyses made and statistical calculations included are clear and could be verified and interpreted by readers. 

Re: Figures and Tables 

Line 148: Should Figure 2 be cited here or at the beginning of this paragraph? Indeed citation of new Figure 1 could be placed in sentence ending on line 110. 

Line 161: End of sentence should cite Table 1 not Figure 1

The clarity of Figure 3 (previously Figure 2)  has improved but data cannot be independently verified based on inconsistences of data presented concerning the presentation of clinical metrics presented in tables verses data interpretation made in text (see above). The final version of this figure should include a larger font size of the gene names as this is still difficult to read. 

Re: Discussion

The discussion also has some issues with some statements made that do not align with data presented such as the following examples that also include other comments:

Lines 244-253: There are BRCA1 variants that recur multiple times as shown in Table, including BRCA1 c.1687 C>T.  As mentioned above the number of BRCA1 recurrent variants needs to re-checked in Table 1.

Lines 267-276: Consider making a comment concerning the observation that 3 carriers of PALB2 c.2257C>T variant were identified and possibility that this PALB2 may also recur in the Romanian population.

Lines 295-297: I suggest that you rephrase “……and HBC risk is controversial as there is a valid proof of an increased cancer risk only for homozygous individuals” to the following “……and HBC risk is controversial as there a higher level of evidence that carriers homozygous for MUTYH pathogenic variants have an increased risk of BC.”

Lines 299-302: Table 1 indicates that there are 5 carriers of MUTYH pathogenic variants in heterozygous state were 3 such individuals (not 5) harboured the c.650 G>A variant. Given this discrepancy, it is not clear which variant was associated with the histological subtype of BC in the sentence that follows. 

Lines 370-373: Perhaps consider adding BRCA1 c.1687C>T and PALB2 c.2247 C>T to this list of variants that recur at least 3 times or more in your study cohort?

Final comments: 

There is a lot of interesting data presented in this study that with clarification would improve the manuscript and be of value to medical geneticists in the field. 

Author Response

Esteemed Editor and Reviewer

We made all corrections suggested by the reviewer no. 1 and uploaded the new document.

We completed all modifications as comments to the text for each of the comments, suggestions, and specific requirements. 

Thank you for still considering our manuscript for publication.

Dr. Catana Andreea

Round 3

Reviewer 1 Report

The found the recent revision of the mansucript to address my questions and concerns.

Minor  comments:

1. Lines 130- 133: Suggest to add for clarity -  "We identified 142 defects in all. Of these, 77 were different pathogenic variants, of which ......"

2. Suggest you reference both Fig 1 and 2 at end of sentence on line 134, as Fig 1 states the number of patients patients with two pathogenic mutations. 

3. Line 269: Edit "BRCA11" to "BRCA1". 

Author Response

Esteemed Editor and Reviewer, we uploaded the manuscript with the minor revisions required. 

Thank you.
